# Multidimensional performance characteristics of youth academy and club soccer players

Ulrikke Norill Kvalvaag[1☉], Hege R. Eriksen[1], Hilde Gundersen[1], Thomas Johansen[2,3], Anne Marte Pensgaard[4], Morten Kristoffersen[1☉*]

1 Department of Sport, Food and Natural Sciences, Western Norway University of Applied Sciences, Bergen, Norway, 2 Norwegian National Advisory Unit on Occupational Rehabilitation, Rauland, Norway, 3 Hernes Occupational Rehabilitation Centre, Hernes, Norway, 4 Institute for Sport and Social Sciences, Norwegian School of Sport Sciences, Oslo, Norway

☉ These authors contributed equally to this work.
* Morten.Kristoffersen@hvl.no

## Abstract

Talent identification and development (TID) in soccer is complex. While physical performance and skeletal age (as an estimate of biological maturation) are well-established in TID research, the combined role of psychological skills and cognitive function remains underexplored. The aim of this study was to examine multidimensional performance characteristics among U14 players by comparing match-selected academy players (n = 20), non-match selected academy players (n = 14), and club players (n = 22). In total 56 players performed 40 m sprint test, Yo-Yo Intermittent Recovery Test Level 1 (IR1), and completed validated self-reported mental skills and motivation questionnaires. Executive functions were assessed using tests from Cambridge Neuropsychological Test Automated Battery (CANTAB) and skeletal age was assessed from left-hand X-ray images. The results showed that match-selected academy players were significantly taller, had higher weight and bone age compared to both club players and non-match-selected academy players. Match-selected academy players had significantly better physical performance and reported stronger mental skills, compared to club players. Club players reported higher levels of amotivation and external regulation compared to both match-selected and non-match-selected academy players, reflecting a lessself-determined motivational profile. Match-selected academy players demonstrated better decision-making, characterized by a significantly lower tendency to take risks to avoid delay and a greater ability to rationally adjust risk compared to club players. No further group differences were found for the executive function tests between groups.

## Introduction

Performance in soccer is shaped by an interaction of physical, technical, tactical, cognitive, psychological, and psychosocial attributes [1–3]. However, the abilities

**Data availability statement:** Due to the sensitive nature of the data and the involvement of minors, the dataset cannot be shared publicly. Data requests may be directed to the Data Protection Officer at Western Norway University of Applied Sciences (HVL) (personvern-afii@hvl.no) and will be considered on a case-by-case basis in accordance with applicable ethical and legal restrictions.

**Funding:** The author(s) received no specific funding for this work.

**Competing interests:** The authors have declared that no competing interests exist.

that are important for performance at a young age do not necessarily affect performance later [4], making talent identification and development (TID) complex and widely debated [5,6]. Talent identification refers to the process of recognizing young athletes based on attributes thought to be related to long-term performance. Talent development involves providing athletes with learning environments that help them realize their potential, while talent selection refers to the ongoing decisions about who progresses to increasingly exclusive stages of the system [7,8]. Although clubs and national federations increasingly attempt to identify players at progressively younger ages, this practice reflects structural and competitive pressure within TID systems rather than evidence that early performance can predict future performance [8,9]. Adolescents development and talent-related characteristics are highly non-linear and varies substantially between individuals, meaning early performance has limited prognostic value [10]. Early identification does not mean we can accurately predict future performance. It simply reflects that players who are noticed early often gain access to better training, coaching, and competitive opportunities [7].

Talent identification and selection evaluations often rely on subjective coach judgments, which sometimes informative [11,12], are also vulnerable to bias and tend to overemphasize short-term performance [9]. This may lead to selection bias due to maturational advantages such as biological maturity and relative age effects [13–16], factors widely recognized as influential determinants in talent identification and selection [5,6,15,17]. Because adolescence is characterized by substantial inter-individual variability in growth and development [18], early maturing athletes often experience temporary advantages in strength, power and speed, contributing to their overrepresentation in talent development programs [17]. Recent research highlights the importance of accurately assessing maturational status, as skeletal age (commonly measured through X-ray imaging [19]), and growth-related factors strongly shape youth performance profiles both in males and female athletes [20–22]. Similarly, the relative age effect (RAE) describes a systematic advantage for athletes born early in the selection years, who often benefit from greater physical and physiological development compared to their younger peers [23–25]. When these factors are not accounted for, there is an increased risk of overemphasizing short-term performance, potentially at the expense of athletes with long-term development potential.

However, maturation represents only one of several factors influencing selection decisions. Coaches' beliefs about the nature of talent influence the characteristics they prioritize during evaluation [26]. Traditional selection models typically rely on objective testing combined with subjective coach evaluation of technical skills, while insufficiently acknowledging the dynamic multidimensional nature of talent [2,27]. Nevertheless, scientific evidence suggests that young soccer players who demonstrate strong physical [28–31], technical [32,33], cognitive [34], and mental [31], abilities either in isolation or in combination, are more likely to succeed in the long term. At the same time, emerging literature highlights additional contributors to youth development, such as nutritional adequacy and training environment, which are often

overlooked in traditional talent models [35]. Looking at both what to identify but also give attention toward how talent can effectively nurture and developed over time might benefit the TID system.

Despite growing interest in psychological and cognitive predictors of talent in soccer, research in this area remains underexplored, particularly in combination [29,36,37]. Psychological factors, such as mental skills and motivation, have been identified as predictors of future performance and are increasingly recognized as important components in talent development [29,38,39]. However, research in this area is often limited by inconsistent methodologies and narrow variable selection [2]. This creates challenges in terms of content validity, whether the battery of tests truly captures the multidi-mensional nature of performance. At the same time, low scores on specific skills (e.g., physical) may mask the possibility for strengths in unmeasured areas (e.g., psychological skills), highlighting the need for broader assessment protocols [2]. Furthermore, soccer is an open-skill sport, characterized by dynamic and unpredictable environments that require athletes to continuously adapt their actions in response to situational demands [34]. Tactical performance in soccer relies heavily on cognitive abilities such as decision-making, attentional-control and problem-solving, all of which have been strongly associated with executive functions (EF) [40].

EF encompass core cognitive processes such as inhibition, working memory, and cognitive flexibility, as well as higher-order skills including decision-making and risk-taking [41]. These abilities support athletes' capacity to regulate behavior and adapt to dynamic match situations [34,41]. Several studies also report that high-performing youth soccer players out-perform their lower-level peers on EF tasks [34,36,42–45]. EF development is closely linked to maturation of the prefrontal cortex, which develops gradually throughout adolescence and into the mid-twenties [46,47].

The different components of EF develop individually at different rates and are influenced by factors such as physical activity, sports participation, and other environmental factors [41]. It remains unclear whether enhanced EF in young high-performing athletes are primarily shaped by maturation and age (nurture) or the amount of sport-specific experience (nature). Although EF have been associated with soccer performance, effect sizes are generally small [48], and the litera-ture is characterized by heterogeneous tasks and inconsistent methods [48,49]. Furthermore, EF is rarely explored within multidimensional test batteries, limiting our understanding of how it relates to other performance variables.

While physical and maturational factors in TID have been extensively documented [2,20,50,51], the combined contri-bution of mental skills, motivation, and executive functions remains relatively underexplored. Therefore, the aim of the present study was to compare match-selected academy players, non-match-selected academy players and club players across physical, maturational, psychological and cognitive domains, using a multidimensional test battery. This may pro-vide insight into how different these attributes relate to athlete development and maturation.

Based on previous research we formulated the following hypothesis:

1.  Match-selected and non-match-selected academy players will show more advanced biological maturation and superior physical performance compared to club players.

2. Match-selected and non-match-selected academy players will outperform club players on mental skills, motivation and executive function task.

## Methods

### Participants

A total of 56 male youth soccer players from Norway participated in this cross-sectional study: 34 of the participants were academy players selected for a national talent development initiative. Of these, 20 players had been selected for match-play participation during the season (age: 13.99 ± 0.22 years), while 14 players had not been selected for match-play participation (age = 13.92 ± 0.36 years). The remaining 22 participants were age-matched club players from a local club (age = 14.0 ± 0.31 years). The academy players trained together once per week as part of the national development

initiative. In addition, they also trained and played matches regularly with their own club. This was organized regionally in collaboration with an elite club that competes in a national league under the "Norwegian Top Football" talent development initiative [52]. Within this structure, a subset of the academy players was selected to represent the club in official matches during the season. Club players serving as a comparison group were recruited from the same geographic area. Not all players completed every component of the test battery due to unrelated minor injury, illness, or participants needing to leave early for personal reasons.

## Procedure

Out of 56 players included in the study, 44 (34 academy players and 10 club players, all born in 2009) were assessed between 20-05-2023 and 28-06-2023, while an additional 12 club players (born in 2010) were assessed during 20-05-2024 and 20-06-2024. At the time of recruitment, the academy players had been part of the national talent development program for five months, corresponding to part of one competitive season. The classification of three groups enabled direct comparison of performance between the three groups as well as examination of factors potentially influencing coaches' match selection decisions.

Testing was conducted over two days, with participants assessed in groups of 10–15 under standardized and controlled conditions. On the first test day, participants completed a questionnaire with self-reported mental skills and motivation, followed by X-ray assessment of skeletal age and anthropometric measurements (height and weight). These tests were consistently scheduled between 15:00 and 17:00. On the second test day, participants completed lab-based executive function testing, followed by a 40-meter sprint test and the Yo-Yo Intermittent Recovery Test Level 1 (IR1). These tests were consistently scheduled between 09:00 and 11:00. Before the physical tests all participants followed the same standardized warm-up, consisting of 15 minutes light jogging followed by three progressive sprints. All test sessions took place at the same locations and were supervised by the same experienced test personnel and trained research assistants to ensure consistency.

## Measurements

**Chronological age, birth quartiles, anthropometrics and skeletal age.** Height and body weight were measured with participants barefoot and in light clothes. Height was measured to the nearest 0.1 cm using a fixed stadiometer (Seca 206, GmBh, UK). Body weight was measured to the nearest 0.1 kg using a calibrated digital scale (Seca 877, GmBh, UK). Participants' birthdates were used to categorize them into relative age quartiles (Q1-Q4), allowing for examination of potential RAE.

All players underwent an X-ray of their left wrist to assess skeletal age, as a validated measurement of biological age and maturational status [53]. Skeletal age was treated as a continuous variable as this approach preserves sensitivity and avoids cut-off points in defining early, average or late maturation [54]. Radiographs were acquired using a Siemens Ysio Max system with the integrated compact FLUORPSPOT imaging system (software version VE10; Siemens Healthineers). The imaging field covered the entire hand in a posterior-anterior projection, plus 3 cm of the distal forearm to capture the epiphyseal plates in both the radius and ulna. Acquisition parameters included a tube-detector distance of 1 m, X-ray energy of 50 kV, and exposure of 1–1.5 mAs, with no subsequent processing or filtering applied. BoneXpert Standalone version 3.4.1.0 (Visiana, 2019) was used to analyze the radiographs, which automatically calculates 8–13 independent skeletal age estimates from different hand bones. This fully automated approach eliminates inter- and intra-observer variability, with the skeletal age assessments based on the Greulich-Pyle methodology [19]. The root mean square error (RMSE) of BoneXpert is estimated to be 0.68 years in males and 0.52 in females [55]. Skeletal age estimates may vary depending on the assessment protocol used, with Greulich-Pyle estimate tending to be lower than Fels estimate [56].

**Intermittent-endurance capacity.** The Yo-Yo intermittent Recovery Test Level 1 was implemented to evaluate the players' intermittent endurance capacity [57]. This test, which requires periods of acceleration and deceleration, has been shown to correlate strongly with match-related physical performance in youth soccer, thereby supporting its construct

validity [58]. In adolescent soccer players, the test has demonstrated excellent reliability, with ICC values ranging from 0.87 to 0.95 [59]. The assessment was conducted in an indoor gymnasium on wooden sports floor, starting at a standardized speed in accordance with established protocols [51,60]. Total meters covered were recorded as the outcome measure.

**40-m linear sprint.** Sprint performance was assessed on an indoor track using an electronic timing system with wall-mounted photoelectric gates (IC Control Track Timer). The first gate was positioned 50 cm above the ground at the starting line, while the gates at 10, 20, 30 and 40 meters were set at a height of 120 cm [61]. Linear sprint 40 meters is considered valid and demonstrates high intra- and inter-day reliability in a soccer context, with reported ICC between 0.87 to 0.99, for similar linear sprint tests [62]. Each player performed three maximal sprints, with 3–4 minutes of rest between runs. Sprints began from standing position, with front foot placed 60 cm behind the first timing gate, starting with no backward movement, in line with previous protocols [51]. The fastest sprint times for all gates were used in the analysis.

**Mental skills.** Self-perceived mental skills were assessed through a validated [63], soccer version of the Norwegian Mental Skill Questionnaire for Athletes (NMS-42) [64], "mental-skill test-football" (MST-f) [65]. NMS-42 has been used for decades by the Sport Psychological Department of the Norwegian Olympic Training Center [64,66]. The questionnaire consists of 42 items, covering six performance-relevant mental dimensions (7 items in each dimension): Self-talk and self-confidence (e.g., "I believe in myself as a soccer player"), Goal setting and motivation (e.g., "I believe in myself, and expect to achieve my goals"), Energy management (e.g., "Too much stress rarely leads to poor performance"), Concentration (e.g., "I am able to maintain strong concentration throughout the whole game"), Match preparation (e.g., One of the reasons I perform consistently well is that I have good pre-game routines"), and Imagery (e.g., "When I use imagery, I use all my senses"). These dimensions reflect cognitive and emotional strategies athletes use to regulate performance. The items were rated on a 10-point Likert scale from 1 = completely agree to 10 = completely disagree. A sum score for each dimension was calculated, with low scores indicating high perceived use and control of the mental dimension.

Internal consistency for the MST-subscales was assessed using Cronbach´s alpha. Reliability was acceptable for Match preparation (α = .83), Imagery (α = .86), Self-talk and self-confidence (α = .88), Concentration (α = .87), and Goal setting and motivation (α = .88). The Energy management subscale showed low internal consistency in its original 7-item form (α = .35). Item 10 showed a negative corrected item-total correlation and was removed; the resulting 6-item score showed α = .51 and was used in all analyses.

**Motivation.** The Norwegian version of Behavioral Regulation in Sport Questionnaire-20 (BRSQ-20) was used to assess the participants' perceived motivation [67]. The BRSQ-20 is grounded in Self-Determination Theory (SDT), which assesses motivation along a continuum of self-determination, ranging from intrinsic motivation to amotivation [68]. The questionnaire included 20 items, covering five dimensions from SDT, with 4 items in each dimension: Intrinsic motivation (e.g., "Because I enjoy it"), Identified regulation (e.g., "Because the benefit of soccer is important to me"), Introjected regulation (e.g., "Because I would feel ashamed if I quit), External regulation (e.g., "Because people push me to play"), and Amotivation (e.g., "But I question why I continue). The BRSQ has been validated in research, specifically in studies involving participants of the same age group as well as in Norwegian soccer players [69]. Responses were recorded on a 7-point Likert scale (1 = not true at all, 4 = neither true nor false, 7 = completely true). A sum score was calculated, with high score indicating higher value on each dimension.

Internal consistency for the BRSQ-20 motivation scales was acceptable for Intrinsic motivation (α = .70), Integrated regulation (α = .77), Introjected regulation (α = .84), and Amotivation (α = .88). External regulation showed a little lower internal consistency (α = .62), but still acceptable.

**Executive functions.** EF were assessed using the Cambridge Neuropsychological Test Automated Battery (CANTAB, version 1.5.704 in 2023 and 1.5.708 in 2024), a validated, highly sensitive, objective and widely used computerized tool for measuring cognitive performance [70–83]. The CANTAB battery was originally developed for use in clinical

populations, particularly older adults [84], but has since been increasingly applied in non-clinical populations [43,85–87]. Five tasks targeting core EF domains (inhibition, working memory, cognitive flexibility) and higher-order EF (decision-making) were included in this study, along with one task assessing basic reaction time. A detailed description of each outcome measure used in the analyses for each cognitive test, including its code, description, and associated cognitive domain, is provided in S1 Table, while validation and reliability are reported in S2 Table. Prior to testing, participants completed a brief self-reported questionnaire designed to assess control variables related to sleep, caffeine intake, gaming habits, self-efficacy and concentration abilities. The tests were administered individually on identical iPads with standardized headphones, using built-in task instructions in English and response registration. To ensure full comprehension, standardized instructions were also provided in Norwegian by the test personnel.

**Reaction time task:** Reaction time was measured using the Reaction Time (RTI) task, assessing attention, response speed and impulsivity. Participants responded to visual stimuli across five locations, pressed and held a button until a yellow dot appeared in one of five circles, prompting them to select the corresponding circle.

**Spatial span test:** Working memory was assessed via the Spatial Span (SSP) test. In this task, nine white squares were displayed on the screen, with a subset of them lighting up in a specific sequence. Participants were required to recall and replicate the sequence by selecting the squares in the correct order. The sequence length increased progressively.

**Intra-extra dimensional set shift:** The Intra-Extra Dimensional Set Shift (IED) is a computerized version based on the Wisconsin Card Sorting Test [88]. The test assesses cognitive flexibility, specifically the ability to learn rules and shift attention between relevant and irrelevant stimulus dimensions. In this task, participants were required to rely on feedback to infer the rule that identified the correct stimulus. The rule shifts after six consecutive correct responses, progressing from simple, single-dimension discrimination (e.g., variations in white line shapes) to more complex, compound stimuli involving both white lines and pink shapes. Initially, rule shifts occur within the same dimension (intra-dimensional shifts) before advancing to shifts between dimensions (extra-dimensional shifts).

**Cambridge gambling task:** Decision-making was assessed using the Cambridge Gambling Task (CGT), which measures risk-taking behavior, delay aversion, and the ability to adjust bets according to outcome probabilities. Participants were presented with a row of ten boxes, colored red and blue in varying proportions across trials, with a yellow token randomly hidden behind one of the boxes. They first selected the color they believed the token was hidden behind, then chose how many points to bet on their decision. The bet value increased or decreased automatically across trials in either an ascending or descending format, requiring participants to decide when to place their bet.

## Ethics statement

All participants and their parents received detailed written information about the research project and procedures prior to participation. Written informed consent, including information about the right to refuse to participate or to withdraw consent to participate at any time without reprisal, was obtained from the participants and their legal guardians prior to data collection. The study was reviewed by Regional Committees for Medical and Health Research Ethics (REK), which concluded that ethical approval was not required. The study was evaluated by the Norwegian Center for Research Data (212585) and conducted in accordance with the Declaration of Helsinki.

## Statistical analysis

Data were analyzed using IBM SPSS Statistics, Version 30.0 (IBM Corp., Armonk, NY, USA). Normality was assessed using the Shapiro-Wilk test, Q-Q plots and box plots. Given the Shapiro-Wilk test's sensitivity in small samples [89], significant deviations from normality ($p < .05$) were further evaluated visually. Where non-normality was detected, both parametric and non-parametric tests were conducted. As results did not differ meaningfully, only parametric outcomes are reported. A one-way analysis of variance (ANOVA) was used to compare match-selected academy players, non-match-selected academy players, and club players across all performance variables. An alpha level of 0.05 was used

to determine statistical significance. Tukey´s post hoc analyses were conducted to identify pairwise group differences. Effect sizes for the ANOVA results were calculated as eta squared ($\eta^2$) for comparisons between the three groups. As there were no statistically significant group differences control variables measuring sleep, gaming, caffeine intake and concentration, these variables were not included in the final analysis.

## Results

### Differences between academy and club players

**Chronological age, birth quartiles, anthropometrics and bone age.** The mean chronological age of the participants was 14.0 ± 0.3 years ($n = 56$), while the mean bone age was 14.1 ± 1.2 years ($n = 53$). No significant differences were observed between the three groups in birth quartile distribution (Q1-Q4) ($F(2,51) = 0.22$, $p = .796$, $\eta^2 = .009$. Academy players selected for match play had significantly higher weight, height, and bone age compared to both club players and academy players not selected for match play, while no group differences were observed in chronological age (Table 1).

**Physical performance.** Significant between-group differences were found for 10 m, 20 m, 30 m, 40 m sprint times, as well as IR1 performance (all $p < .01$). Post hoc comparisons showed that match-selected academy players outperformed club players across all sprint distances and IR1, and were also faster than non-match-selected academy players at 30 m and 40 m sprint (Table 2).

**Mental skills.** The match-selected academy players demonstrated significantly better scores across all self-reported mental skills compared to the club players (all $p < .05$). Additionally, the non-match-selected academy players reported significantly better scores in self-talk and confidence, concentration, and goal setting and motivation compared to club players (Table 3).

**Motivation.** Significant between-group differences were found for identified regulation, external regulation and a-motivation (all $p < .05$, Table 4). Post hoc comparisons showed that club players scored higher than match-selected academy players on identified regulation. Club players also scored higher (less favourable) than both non-match-selected and match-selected academy players on external regulation and a-motivation. No significant differences were observed for Intrinsic motivation or Introjected regulation (Table 4).

**Executive functions.** Significant between-group differences were shown for delay aversion total and for risk adjustment total merged (Table 5). Post hoc analyses for both measures showed that selected academy players scored

**Table 1. Comparison of chronological age, bone age, height and weight between match-selected academy players, non-match-selected academy players and club players, using one-way ANOVA with post hoc tests ($n = 56$). Values are presented as mean (SD).**

| | F (df1, df2) | p-value | $\eta^2$ | Match-selected academy players (MS) n = 20 | Non-match-selected academy players (NMS) n = 14 | Club players (CP) n = 22 | Post hoc |
|---|---|---|---|---|---|---|---|
| Chronological age | 0.57 (2, 51) | .570 | .022 | 13.99 (0.22) | 13.92 (0.36) | 14.03 (0.33) | – |
| Bone age[a] | 7.49 (2, 50) | .001 | .230 | 14.75 (1.04) | 13.39 (0.97) | 13.89 (1.06) | MS > NMS, MS > CP |
| Height[b] (cm) | 8.31 (2, 52) | <.001 | .242 | 174.30 (6.20) | 163.11 (11.40) | 166.45 (7.70) | MS > NMS, MS > CP |
| Weight[c] (kg) | 7.73 (2, 52) | .001 | .229 | 61.30 (7.79) | 51.47 (10.09) | 52.38 (7.84) | MS > NMS, MS > CP |

[a] Bone age data available for $n = 13$ non-match-selected academy players and $n = 20$ club players.

[b] Height available for $n = 21$ club players.

[c] Weight available for $n = 21$ club players.

**Table 2. Comparison of IR1 and 40-meter linear sprint between match-selected academy players, non-match-selected academy players, and club players, using one-way ANOVA with post hoc tests (_n_ = 52). Values are presented as mean (SD).**

| Variables | F (df1, df2) | _p_-value | η² | Match-selected academy players (MS) _n_ = 20 | Non-match-selected academy players (NMS) _n_ = 10 | Club players (CP) _n_ = 22 | Post hoc |
|---|---|---|---|---|---|---|---|
| IR1 (meters) | 6.72 (2,45) | .003 | .230 | 1332 (92)[a] | 1137 (118) | 972 (341) | MS > CP |
| 10 m linear sprint (sec) | 8.20 (2,49) | <.001 | .251 | 1.70 (0.06) | 1.75 (0.05) | 1.80 (0.09) | MS > CP |
| 20 m linear sprint (sec) | 8.22 (2,49) | <.001 | .251 | 3.05 (0.12) | 3.18 (0.11) | 3.24 (0.19) | MS > CP |
| 30 m linear sprint (sec) | 6.27 (2,49) | .004 | .204 | 4.35 (0.20) | 4.55 (0.18) | 4.61 (0.28) | MS > NMS, MS > CP |
| 40 m linear sprint (sec) | 7.89 (2,49) | .001 | .244 | 5.60 (0.24) | 5.92 (0.27) | 5.97 (0.38) | MS > NMS, MS > CP |

[a]IR1 data available for _n_ = 16 match-selected academy players.

Abbreviations: IR1, Intermittent Running Test Level 1.

**Table 3. Comparison of mental skills between match-selected academy players, non-match-selected academy players and club players, using one-way ANOVA with post hoc tests (_n_ = 55). Values are presented as mean (SD).**

| Variables | F (df1, df2) | _p_-value | η² | Match-selected academy players (MS) _n_ = 20 | Non-match-selected academy players (NMS) _n_ = 14 | Club players (CP) _n_ = 21 | Post hoc |
|---|---|---|---|---|---|---|---|
| Match preparation[a] | 7.12 (2,52) | .002 | .215 | 20.97 (9.00) | 24.36 (6.40) | 30.14 (7.55) | CP > MS |
| Imagery[a] | 6.13 (2,52) | .004 | .191 | 20.88 (7.83) | 22.93 (11.20) | 29.71 (6.55) | CP > MS |
| Self-talk and Confidence[a] | 9.67 (2,52) | <.001 | .696 | 18.49 (6.10) | 21.71 (9.86) | 29.62 (8.97) | CP > MS, CP > NMS |
| Energy management[a] | 4.43 (2,52) | .017 | .146 | 27.11 (6.86) | 30.33 (5.97) | 33.44 (7.26) | CP > MS |
| Concentration[a] | 13.4 (2,52) | <.001 | .340 | 17.54 (6.50) | 22.01 (8.09) | 29.14 (7.28) | CP > MS, CP > NMS |
| Goal setting and Motivation[a] | 23.76 (2,52) | <.001 | .477 | 15.07 (5.36) | 26.86(9.18) | 29.62 (8.97) | CP > MS |

[a]Lower score indicate favorable outcome on this measure.

significantly better on these outcomes compared to club players. No other significant differences were observed across any of the other EF tasks and the RTI task.

## Discussion

The aim of the study was to compare match-selected academy players and non-match-selected academy players and local club male U14 soccer players in physical, psychological, cognitive, and maturational factors. Match-selected academy players were taller, had higher weight and skeletal age compared to both club players and non-match-selected academy players. Match-selected academy players outperformed club players in physical performance and reported higher mental skills and a more self-determinant motivational profile compared to club players. Match-selected academy players also demonstrated better decision-making, characterized by a significantly lower tendency to take risks to avoid delay and a greater ability to rationally adjust risk compared to club players. No further group differences were found for the executive function tasks between groups.

### Physical and maturational factors

There were no significant differences in chronological age or birth quarter distribution (Q1-Q4) between the three groups. However, significant differences were found in height, weight, and skeletal age, with match-selected academy players being

**Table 4. Comparison of motivation between match-selected academy players, non-match-selected academy players and club players, using one-way ANOVA with post hoc tests (*n* = 55). Values are presented as mean (SD).**

| Variables | F (df1, df2) | p-value | $\eta^2$ | Match-selected academy players (MS) *n* = 20 | Non-match-selected academy players (NMS) *n* = 14 | Club players (CP) *n* = 21 | Post hoc |
|---|---|---|---|---|---|---|---|
| Intrinsic motivation[b] | 1.38 (2,52) | .262 | .050 | 6.49 (0.64) | 6.54 (0.57) | 6.17 (0.91) | – |
| Identified regulation[b] | 3.81 (2,52) | .029 | .128 | 1.38 (0.67) | 1.61 (1.02) | 2.10 (0.87) | CP > MS |
| Introjected regulation[a] | 0.35 (2,52) | .708 | .013 | 2.09 (1.62) | 1.96 (1.16) | 2.35 (1.29) | – |
| External regulation[a] | 5.14 (2,52) | .009 | .165 | 3.93 (1.23) | 3.90 (1.29) | 4.93 (0.93) | CP > MS, CP > MS |
| A-motivation[a] | 11.1 (2,52) | <.001 | .299 | 1.16 (0.18) | 1.29 (0.45) | 2.33 (1.32) | CP > NMS, CP > NMS |

[a] Lower score indicate favorable outcome on this measure.

[b] Higher score indicate favorable outcome on this measure.

**Table 5. Comparison of executive functions between match-selected academy players, non-match-selected academy players and club players, using one-way ANOVA with post hoc tests (*n* = 53). Values are presented as mean (SD).**

| | F (df1, df2) | p-value | $\eta^2$ | Match-selected academy players (MS) *n*=19 | Non-match-selected academy players (NMS) *n*=14 | Club players *n*=20 (CP) | Post hoc |
|---|---|---|---|---|---|---|---|
| **Reaction Time Task (msec)** | | | | | | | |
| Median movement time | 1.68 (2,50) | .196 | .063 | 146.78 (25.67) | 165.32 (34.11) | 151.67 (28.74) | – |
| Median reaction time[a] | 0.03 (2,50) | .971 | .001 | 355.63 (27.50) | 357.00 (4.68) | 358.35 (32.67) | – |
| **Spatial Span Task (Working memory)** | | | | | | | |
| Span Length reached [b] | 1.06 (2,50) | .351 | .041 | 7.47 (1.02) | 7.14 (1.02) | 6.95 (1.27) | – |
| **Stop Signal Task (msec) (Inhibition)** | | | | | | | |
| Stop Signal reaction time[a] | 2.12 (2,50) | .130 | .078 | 346.76(124.89) | 313.76 (98.45) | 274.95 (99.18) | – |
| **Intra-Extra Dimensional Task (Cognitive flexibility)** | | | | | | | |
| Errors, extra-dimensional shifts[a] | 0.32 (2,50) | .721 | .015 | 12.24 (10.53) | 9.42 (9.60) | 10.06 (10.02) | – |
| Total errors (stage adjusted) [a] | 0.30 (2,50) | .737 | .012 | 38.58 (29.01) | 38.57 (31.31) | 32.05 (28.09) | – |
| **Cambridge Gambling Taks (Decision-making under risk)** | | | | | | | |
| Delay aversion total [a] | 3.45 (2,50) | .039 | .121 | 0.18 (0.09) | 0.26 (0.13) | 0.29 (0.14) | CP > MS |
| Risk adjustment merged [b] | 1.60 (2,50) | .210 | .061 | 1.03 (0.61) | 1.26 (1.10) | 1.49 (0.69) | – |
| Risk adjustment total merged [b] | 4.38 (2,50) | .018 | .149 | 0.71 (0.15) | 0.62 (0.78) | 0.59 (0.12) | MS > CP |

[a] Lower score indicate favorable outcome on this measure.

[b] Higher score indicate favorable outcome on this measure.

taller, heavier, and more biologically mature compared to both non-match-selected academy players and club players. This supports previous studies documenting the overrepresentation of early-maturing players in soccer academies [50,90]. However, the current study also demonstrates that maturity-related differences can be observed within the academy environment itself. Since no significant differences were found in birth quarter distribution (Q1-Q4), this reinforce the conceptual distinction between RAE and skeletal age, emphasizing the importance of measuring these constructs separately [17].

The physical performance tests showed that match-selected academy players outperformed club players across all sprint distances (10–40 m) and IR1 performance. They also performed better than non-match-selected academy players at the 30- and 40- meter sprint. This support previous findings that higher level youth players exhibit superior speed and intermittent endurance capacities [51]. However, in this sample it remains unclear whether these differences were already present prior to academy selection. We do not know if some of the differences reflect advantages of being more biologically mature or if players taking part in a structured academy environment develop better speed and endurance. This raises important questions regarding in which degree superior physical performance in youth players serves as an indicator of inherent talent or if it primarily results from maturation, differences in training exposure, developmental opportunities, or familiarization with physical testing [91].

Notably, the differences in skeletal age and sprint performance between match-selected and non-match-selected academy players may be a result of coaches' match selection decisions. These findings highlight that, even within a pre-selected academy cohort, biological maturity remains a significant determinant of match selection. While such selection criteria may optimize short-term performance, they carry the risk that coaches overlook late-maturing players with higher long-term potential. Too much focus on short-term results such as match winning during vulnerable developmental phases, like maturational shifts, may also increase the risk of injury or long-term dropout due to physical overload [92].

### Mental skills and motivation

Club players scored less favorable than both academy groups in all mental skills domains, including match preparation, imagery, self-talk and confidence, concentration, and goal setting and motivation. Despite participating in academy training only once per week, even limited exposure to a structured environment can potentially increase psychological skills, through professional coaching and higher level of competition. Furthermore, mental skills measured in the study have all been shown to be important for player development and future success in soccer [3,39,93,94]. No differences in mental skills or motivation were found between match-selected and non-match-selected academy players. This may imply that players selected for the academy possess stronger mental skills, or that participating in a structured training program provides positive psychological benefits. Club players further reported higher levels of identified regulation compared to match-selected academy players, but also higher levels of external regulation and amotivation compared to both academy groups. Although club players reported significantly higher levels of identified regulation, this does not necessarily reflect a robust motivational profile. While identified regulation is considered a relatively autonomous form of motivation, indicating that players recognize the personal value of participating in soccer, it may still be driven by performance-related goals and external expectations [95]. This is highly relevant given that club players also reported higher levels of external regulation and amotivation, suggesting that their motivation may be influenced by external pressures, such as selection aspirations, social approval, or fair of failure. Such a profile, specifically externally driven motivation, is less stable over time and increases the vulnerability to dropout or burnout over time [96,97]. Findings from this cross-sectional study highlights the potential value of incorporating psychological assessments such as mental skills and motivation, into TID framework and research. However, longitudinal research is needed to determine how these attributes in combination relate to long-term development and how they potentially change over time.

### Executive functions

Match-selected academy players demonstrated a greater tendency to take risks when the probability of success was high and displayed lower delay aversion compared to club players. These patterns suggest a more strategic decision-making profile, characterized by calculated risk taking and a greater willingness to wait for optimal opportunities rather than making hasty decisions. While risk-taking may be considered advantageous in soccer, particularly in situations requiring bold and strategic decision-making, such as complex passing, last minute inhibition of passes, or goal-scoring attempts, its benefit is context-dependent and difficult to generalize. Furthermore, no significant differences were identified across

the other EF tasks. This is in contrast with the majority of existing literature, which generally reports superior performance among elite youth athletes' on both core and higher order executive function tasks compared to their non-elite peers [34,43–45].

Several explanations may account for the overall lack of group differences in EF performance in this study. First, EF is highly sensitive to situational factors such as sleep quality, stress, and motivation, which may mask group-level differences [98–100]. Even though we measured these cofactors, no differences emerged in self-perceived levels across the groups. However, self-reported measures may not capture the acute and individual variability in these states. Second, the psychometric properties of many commonly used EF tasks, including those in the CANTAB battery, has been questioned in high-performing, non-clinical youth populations, particularly due to its limited ecological alignment with sport-specific cognitive demands [101]. While the CANTAB battery is validated for use in children and adults, there is a notable lack of research examining its applicability in adolescents' athletic population [71,102–104].

Taken together, the lack of group differences in EF should not be interpreted as evidence that such differences do not exist. Rather, the findings may reflect limitations in current EF assessment tools, the limited intra-individual sensitivity of the measures used, sample characteristic or the developmental stage of the participants. This highlights the need for more ecologically valid and developmentally appropriate EF assessments in youth sports, ideally explored through longitudinal research.

## Strengths and limitations

A key strength of this study is the multidimensional approach, incorporating psychological, cognitive, physical, and maturational factors. The inclusion of multiple constructs of EF tasks and mental skills and motivation dimensions, allows for a more comprehensive exploration of factors that may be associated with selection status and potential indicators of players development in youth soccer. The broad dimension of measurement allows for a more comprehensive understanding of individual differences that may contribute to talent development.

This study also has several limitations. The relatively small sample size limits statistical power and generalizability. The academy players had participated in the talent development program for five months prior to testing, which introduces uncertainty regarding whether observed differences reflect selection effects or developmental changes. The cross-sectional design prevents causal conclusions about long-term developmental trajectories. Additionally, the high intra-individual variability and developmental sensitivity of EF may reduce the likelihood of detecting group differences in cross-sectional design [101,105]. Future research should therefore include longitudinal designs and assess measurement reliability within the target population. Lastly, this study did not include soccer-specific tests or measured sociological factors, both of which may significantly impact performance.

## Practical implications

Taken together, at U14 these test results are best interpreted as developmental snapshot, not as stable indicators of long-term potential. Physical test outcomes should therefore be interpreted in light of biological maturation to avoid ranking players on scores that mainly reflect differences in growth. This will help provide a fair foundation for individual development. Selection processes should remain flexible and reviewed regularly, ensuring that late matures are supported through this process.

Psychological profiles should be used to guide individual development plans, not to justify selection or deselection as differences may be an effect of current selection status or other environmental and social factors. Supportive coaching behavior focusing on players motivation and mental skills may strengthen more self-determinate motivation and may reduce the risk of dropout. EF testing should be viewed as supplementary tools to support cognitive development and should be viewed in line with the maturation of the pre-frontal cortex.

## Future directions

Future progress in this field requires longitudinal multidimensional research with repeated measurements of maturation status, physical performance, psychological skills, motivation, and cognitive assessments (both laboratory and soccer specific). Including training exposure and load, nutritional, contextual and psychological factors (coach behaviors, climate, parental pressure) and health markers, could clarify how these factors are connected and shape development. Using longitudinal models can help separate true developmental changes from selection effects, giving a clearer picture of how different factors develop together over time. This may contribute to more informed and less premature approaches to talent identification and selection.

## Conclusion

Match-selected academy players were taller, heavier, and more biologically mature than both club players and non-match-selected academy peers. They outperformed club players across all sprint distances and in the Yo-Yo IR1 test, and were faster than non-match-selected players over 30–40 m sprints. Academy players also reported higher mental skills and a more self-determined motivational profile, with no differences between the two academy groups. These findings suggest maturity advantages persist within pre-selected cohorts and may affect match selection.

For practice this highlights the importance of looking at both physical, psychological and cognitive assessments in relation to biological maturation. Integrating psychological assessments into the TID framework could add potential value. It also highlights the importance of having focus on mental skills and motivation at all levels to reduce the risk of drop-out. However, longitudinal research is needed to determine how these attributes relate to long-term development and how they evolve. Standard lab-based EF tests showed limited sensitivity, highlighting the need for more ecologically valid assessments of executive functioning in youth soccer

## Supporting information

**S1 Table. Cognitive outcome measures.**
(DOCX)

**S2 Table. Reliability and validity of measurement instruments used in the study.**
(DOCX)

## Acknowledgments

We would like to express our gratitude to all the players who participated in this study, as well as the soccer clubs involved for their collaboration and support throughout the data collection process. We would also thank the test leaders and research assistants for their invaluable contribution and conducting the testing sessions under standardized conditions.

## Author contributions

**Conceptualization:** Ulrikke Norill Kvalvaag, Hege R. Eriksen, Hilde Gundersen, Thomas Johansen, Anne Marte Pensgaard, Morten Kristoffersen.

**Data curation:** Ulrikke Norill Kvalvaag, Hege R. Eriksen, Morten Kristoffersen.

**Formal analysis:** Ulrikke Norill Kvalvaag, Hege R. Eriksen, Thomas Johansen, Morten Kristoffersen.

**Funding acquisition:** Hege R. Eriksen, Morten Kristoffersen.

**Investigation:** Ulrikke Norill Kvalvaag, Hege R. Eriksen, Morten Kristoffersen.

**Methodology:** Ulrikke Norill Kvalvaag, Hege R. Eriksen, Hilde Gundersen, Thomas Johansen, Anne Marte Pensgaard, Morten Kristoffersen.

 

**Project administration:** Ulrikke Norill Kvalvaag, Morten Kristoffersen.

**Resources:** Ulrikke Norill Kvalvaag, Hege R. Eriksen, Hilde Gundersen, Thomas Johansen, Anne Marte Pensgaard, Morten Kristoffersen.

**Supervision:** Ulrikke Norill Kvalvaag, Hege R. Eriksen, Morten Kristoffersen.

**Validation:** Ulrikke Norill Kvalvaag, Hege R. Eriksen, Hilde Gundersen, Thomas Johansen, Anne Marte Pensgaard, Morten Kristoffersen.

**Visualization:** Ulrikke Norill Kvalvaag, Morten Kristoffersen.

**Writing – original draft:** Ulrikke Norill Kvalvaag.

**Writing – review & editing:** Ulrikke Norill Kvalvaag, Hege R. Eriksen, Hilde Gundersen, Thomas Johansen, Anne Marte Pensgaard, Morten Kristoffersen.

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
