## [Decision Letter · Decision Letter 0]

27 Jan 2026

PONE-D-25-64451Multidimensional Performance Characteristics of Youth Academy and Club Soccer PlayersPLOS One

Dear Dr. Norill Kvalvaag,

Thank you for submitting your manuscript to PLOS ONE. After careful consideration, we feel that it has merit but does not fully meet PLOS ONE’s publication criteria as it currently stands. Therefore, we invite you to submit a revised version of the manuscript that addresses the points raised during the review process.

We look forward to receiving your revised manuscript.

Kind regards,

Julio Alejandro Henriques Castro da Costa

Academic Editor

PLOS One

Journal Requirements:

2. In the online submission form you indicate that your data is not available for proprietary reasons and have provided a contact point for accessing this data. Please note that your current contact point is a co-author on this manuscript. According to our Data Policy, the contact point must not be an author on the manuscript and must be an institutional contact, ideally not an individual. Please revise your data statement to a non-author institutional point of contact, such as a data access or ethics committee, and send this to us via return email. Please also include contact information for the third party organization, and please include the full citation of where the data can be found.

Reviewers' comments:

Reviewer's Responses to Questions

**Comments to the Author**

1. Is the manuscript technically sound, and do the data support the conclusions?

Reviewer #1: Yes

Reviewer #2: Yes

2. Has the statistical analysis been performed appropriately and rigorously? 

Reviewer #1: Yes

Reviewer #2: Yes

3. Have the authors made all data underlying the findings in their manuscript fully available?

Reviewer #1: Yes

Reviewer #2: Yes

4. Is the manuscript presented in an intelligible fashion and written in standard English?

Reviewer #1: Yes

Reviewer #2: Yes

5. Review Comments to the Author

Reviewer #1: The purpose of this study was to determine multidimensional performance characteristics among U14 players by comparing match-selected academy players (n=20), non-match selected academy players (n=14), and club players (n=22). In total 56 players performed 40 m sprint test, Yo-Yo Intermittent Recovery Test Level 1, and completed validated self-reported mental skills and motivation questionnaires. Executive functions were measured via tests from Cambridge Neuropsychological Test Automated Battery (CANTAB) and biological age was calculated using bone age assessed from left hand X-ray images.

The results indicated that match-selected academy players were significantly taller, had higher weight and bone age compared to both club players and non-match-selected academy players. In addition, match-selected academy players had significantly better physical performance and reported stronger mental skills, compared to club players. Moreover, club players reported higher levels of a motivation and external-regulation compared to both match-selected- and non-match-selected academy players, reflecting a less-self determinate motivational profile. In conclusion, match-selected academy players demonstrated better decision-making, characterized by a significantly lower tendency to take risks to avoid delay and a greater ability to rationally adjust risk compared to club players. Finally, no further group differences were found for the executive function tests between groups

Thank you for the opportunity to review this interesting and timely manuscript. The topic is appropriate and important, as talent identification/selection in soccer remains a current priority in both research and practice, while also being debated in the literature. Overall, the manuscript has a strong introduction and a clearly stated problem; however, several areas require improvement before the study can be strengthened and its contribution made clearer.

Major points for revision

1. Hypotheses (clarity and presentation)

o The hypotheses should be stated more explicitly. Please present the study hypotheses in a clear, direct format (ideally numbered or listed) so the reader can easily understand what was tested and how the analyses relate to the research aims.

2. Reliability and validity reporting

o While the selected tests and measurements appear appropriate, the manuscript needs clearer reporting of reliability and validity for all instruments and assessments used.

o Please add a concise section (or table) summarizing reliability/validity evidence for each test/measurement (e.g., test–retest reliability, inter-rater reliability if applicable, internal consistency when relevant, and validity evidence). If reliability was assessed in the current sample, those results should be reported; if not, provide strong supporting citations and justify why those measures are appropriate for the population and context.

3. Literature review (update and scope)

o The discussion and conclusion would benefit from incorporating more recent literature (2023 and above) to reflect current debates and developments in soccer talent identification.

o In addition, I recommend strengthening the literature review by including a focused section on key problems/challenges in talent identification in football, such as (as relevant to the manuscript): relative age effects, maturation status and growth-related bias, contextual and environmental influences, position-specific demands, selection biases, and the limitations of predicting long-term performance from early testing.

4. Discussion and conclusion (strength, practical applications, future directions)

o The conclusion section should be strengthened to provide clearer practical applications for coaches, academies, and practitioners, aligned with the study findings.

o Please also expand the future research directions, including what additional variables, methods, samples, or longitudinal approaches might address limitations in the current study and improve talent identification models.

I look forward to reviewing a revised version of this interesting and timely manuscript. Thank you.

Reviewer #2: Dear Authors,

I appreciate the opportunity to review the study titled “Multidimensional Performance Characteristics of Youth Academy and Club Soccer Players.” Congratulations on your efforts to present this important comparison of soccer players across different competitive levels. To enhance the content of the paper, please consider my following suggestions, comments, and questions:

[1] P3 L36-38: The idea presented contradicts the initial assertion. You state, “First, to ensure that the most promising talent reaches their full potential, clubs and national federations allocate substantial resources to identifying and nurturing players as early as possible.” If the dynamics of sports change in adulthood, why is there an emphasis on early identification?

[2] P3 L38: Please clarify the term "early talent detection." What does it specifically entail?

[3] P3 L42: While the authors address maturation and the relative age effect, it is important to recognize that other factors influence coach selection. Speculation should be balanced with these additional considerations.

[4] P3 L33-P42: Recent references relevant to youth soccer players could enhance the introduction section. Consider including the following:

Martinho, D. V., Gonzalo-Skok, O., Chamari, K., Field, A., Clemente, F. M., & Rebelo, A. et al. (2026). Nutrition as a missing piece in the development of youth male soccer players: a scoping review and future directions. Biology of Sport, 43(1), 291-317.

Martinho, D. V., Coelho-E-Silva, M. J., Gonçalves Santos, J., Oliveira, T. G., Minderico, C. S., Seabra, A., Valente-Dos-Santos, J., Sherar, L. B., & Malina, R. M. (2023). Body Size, Fatness and Skeletal Age in Female Youth Soccer Players. International Journal of Sports Medicine, 44(10), 711–719. https://doi.org/10.1055/a-1686-4563

[5] P3 L49: It may be more accurate to refer to "skeletal age" rather than "bone age."

[6] P3 L50: The term "physiological development" is incorrect; please revise accordingly.

[7] P4 L55-58: Ensure consistent terminology throughout the text. "Identification," "selection," and "development" represent different concepts. Refer to the following sources for clarity:

Till, K., & Baker, J. (2020). Challenges and [Possible] Solutions to Optimizing Talent Identification and Development in Sport. Frontiers in Psychology, 11, 664. https://doi.org/10.3389/fpsyg.2020.00664

Mendes, D., Martinho, D. V., Travassos, B., Gouveia, É. R., Saavedra, N. O., & Sarmento, H. (2025). Talent Selection in Portuguese National Futsal Teams. Journal of Human Kinetics, 99, 253–261. https://doi.org/10.5114/jhk/199379

[8] Introduction Length: The introduction is overly lengthy. Consider abbreviating the last two paragraphs leading up to the study's aim.

[9] P7 L142: How do you categorize players by maturational status? The authors represent skeletal age as a continuous variable, rather than categorical.

[10] Please discuss the data quality related to skeletal age assessment, including citations from other studies to support your findings.

[11] P8 L149-150: The reduction of variability in the Greulich-Pyle method has been noted in female soccer, as documented in the following study:

Martinho, D. V., Coelho-E-Silva, M. J., Valente-Dos-Santos, J., Minderico, C., Oliveira, T. G., Rodrigues, I., Conde, J., Sherar, L. B., & Malina, R. M. (2022). Assessment of skeletal age in youth female soccer players: Agreement between Greulich-Pyle and Fels protocols. American Journal of Human Biology, 34(1), e23591. https://doi.org/10.1002/ajhb.23591

[12] Tables should not use labels A, B, and C. Please replace these with "academy players," "non-match academy players," and "club players."

[13] Consider adding some tables as supplementary material and designing figures for the main results to present more objective findings.

6. PLOS authors have the option to publish the peer review history of their article (what does this mean?). If published, this will include your full peer review and any attached files.

Reviewer #1: **Yes:** Ferman Konukman

Reviewer #2: No

---

## [Author Response · Author response to Decision Letter 1]

27 Mar 2026

Thank you for the opportunity to revise the manuscript. Please see the uploaded file “Response to Reviewers” for our detailed point-by-point responses to the comments from the academic editor and reviewers. We have revised the manuscript accordingly, and a tracked-changes version has been included with the submission.

---

## [Decision Letter · Decision Letter 1]

20 Apr 2026

Multidimensional Performance Characteristics of Youth Academy and Club Soccer Players

PONE-D-25-64451R1

Dear Dr. Norill Kvalvaag,

We’re pleased to inform you that your manuscript has been judged scientifically suitable for publication and will be formally accepted for publication once it meets all outstanding technical requirements.

Kind regards,

Julio Alejandro Henriques Castro da Costa

Academic Editor

PLOS One

Additional Editor Comments (optional):

Reviewers' comments:

Reviewer's Responses to Questions

**Comments to the Author**

1. If the authors have adequately addressed your comments raised in a previous round of review and you feel that this manuscript is now acceptable for publication, you may indicate that here to bypass the “Comments to the Author” section, enter your conflict of interest statement in the “Confidential to Editor” section, and submit your "Accept" recommendation.

Reviewer #1: All comments have been addressed

2. Is the manuscript technically sound, and do the data support the conclusions?

Reviewer #1: Yes

3. Has the statistical analysis been performed appropriately and rigorously? 

Reviewer #1: Yes

4. Have the authors made all data underlying the findings in their manuscript fully available?

Reviewer #1: Yes

5. Is the manuscript presented in an intelligible fashion and written in standard English?

Reviewer #1: Yes

6. Review Comments to the Author

Reviewer #1: I believe that the current edited version of this manuscript is acceptable. All my questions were answered in detail. Thank you.

7. PLOS authors have the option to publish the peer review history of their article (what does this mean?). If published, this will include your full peer review and any attached files.

Reviewer #1: **Yes:** Ferman Konukman

---

## [Editor Report · Acceptance letter]

PONE-D-25-64451R1

PLOS One

Dear Dr. Norill Kvalvaag,

I'm pleased to inform you that your manuscript has been deemed suitable for publication in PLOS One. Congratulations! Your manuscript is now being handed over to our production team.

Kind regards,

on behalf of

Dr. Julio Alejandro Henriques Castro da Costa

Academic Editor

PLOS One